# Whole Grape Pomace Flour as Nutritive Ingredient for Enriched Durum Wheat Pasta with Bioactive Potential

**DOI:** 10.3390/foods12132593

**Published:** 2023-07-04

**Authors:** Carmela Gerardi, Leone D’Amico, Miriana Durante, Maria Tufariello, Giovanna Giovinazzo

**Affiliations:** Institute of Sciences of Food Production, National Research Council of Italy, Via Lecce-Monteroni, 73100 Lecce, Italy; carmela.gerardi@ispa.cnr.it (C.G.);

**Keywords:** grape pomace, bioactive molecules, antioxidants, functional foods, byproduct reuse, fortified pasta

## Abstract

In this study, grape pomace is used as an ingredient to fortify pasta. The grape pomace phenolic component is highly accessible and available for metabolization in the human gut. Hence, grape pomace can be exploited as a source of polyphenols and fiber for sustainable and dietary beneficial food production. Analyses of soluble and bound phenols and volatile compounds in raw and cooked pasta were performed. In the uncooked pasta fortified with pomace, the content of soluble and bound phenolic molecules increased significantly. During the cooking process, the bound phenols were lost, while the soluble phenols doubled. The whole grape pomace flour as a pasta ingredient increased the fiber component by at least double, increased the soluble polyphenol component by at least 10 times, and doubled the isoprenoids (toco-chromanols and carotenoids) while maintaining the unaltered fatty acid content after cooking. In accordance with the polyphenol content, antioxidant activity resulted higher than that of the control pasta. Analysis of volatile compounds in fortified pasta, both uncooked and cooked, indicated an improvement in aromatic profile when compared to the control pasta. Our results show that durum wheat pasta fortified with whole pomace flour has bioactive potential for the reuse of food industry byproducts.

## 1. Introduction

Pasta is among the most consumed foods worldwide, is part of the Mediterranean diet, and has a high level of acceptability regardless of the social class to which the consumer belongs. Italy is among the 10 countries with the highest consumption per capita [1]. Because it consists basically of wheat flour, pasta can be considered a food with a high energy value, rich in complex carbohydrates; however, it has deficiencies from the nutritional point of view in relation to vitamins, minerals, and fibers [2].

Due to its food importance, accessibility, and low value, in addition to being one of the most consumed products on a daily basis, pasta can be considered ideal for adding important ingredients to the diet. The search for alternative sources of healthier and more balanced food is increasing daily, leading the food industry to seek new technological processes that enable improved nutrition, and that do not have high processing costs, in order for them to reach all economic classes of consumers [3].

In order to improve the nutritional value of foods, three approaches can be considered: (1) improving an existing formulation by adding and/or replacing certain ingredients; (2) developing new enriched products using raw materials previously selected as a source of essential nutrients; (3) evaluating the volatile fraction in final products. Each component to be added is characterized as an essential nutrient, providing healthiness to the consumer. In this way, the food can be classified as functional, which by definition is any ingredient in a food capable of providing health benefits, including the prevention and treatment of diseases [4].

Therefore, the growing market for functional products, coupled with consumer interest in disease prevention, has driven the food industry to encourage direct research into the development and acceptability of new products in this regard. In this context, functional foods are gaining more and more success due to their possible health benefits [5].

The food industry produces byproducts that may be considered an opportunity for sustainable valorization in order to minimize waste and ensure environmental defense [6]. The reuse of winery byproducts is of interest to industry, research, and consumers. In order to valorize byproduct components, their inclusion in food products can increase their nutritional value. Recently, some reports investigating the comprehensive utilization of wine-making byproducts have been published [7,8]. However, the researchers chiefly focused on the recovery of some functional components from grape pomace (GP), such as total polyphenols, tocopherols, anthocyanins, and flavan-3-ols [7,8,9]. Only a few reports studied the investigation of the application of whole grape pomace flour (GPF) as an ingredient in foods [10]. Therefore, studies on the physicochemical and functional properties of pasta with GP powder added as a functional ingredient [11,12,13] are encouraged.

About 50% of the GP is composed of grape peels, depending on the grape variety, as well as pedo-climatic and winemaking conditions [8,12,14]. GP is a source of polyphenolic compounds and dietary fiber [15,16,17], components that can exert antioxidant and antimicrobial action [18,19]. The most important GP components are anthocyanins, hydroxycinnamic acids, catechins, and flavonols, which can determine the inhibition of oxidative processes of low-density lipoproteins [7,9]. Several health difficulties such as cardiovascular diseases, stroke, and some cancer types can be prevented by an adequate intake of bioactive compounds through fruits and vegetable use [20,21]. The potential activities of polyphenols and dietary fiber in food quality and safety preservation were demonstrated in some previous studies [13,22,23]. GP contains up to 60% dietary fiber, the insoluble fraction prevailing, followed by sugars, which can total up to 70%, depending on the vinery process applied [7,24]. On the other hand, fortification of food products with grape byproducts led to a nutritional value increase due to the intake of fiber and polyphenolics with antioxidant activity [16,25]. Dietary fiber plays an essential role in human health, such as the improvement in gastrointestinal activity and a reduction in the glycemic response and cholesterol levels in the blood [13]. For this reason, since the recommended intake of fiber is about 25–30 g per day, it could be necessary to take alternative sources of dietary fiber [22]. Studies have described the production of functional bakery products or pasta by incorporating grape pomace skin powder [12,24,25,26]. Grape byproducts can be used as additive ingredients in functionalized pasta with positive effects, such as increased total phenolic content and antioxidant activity, and reduced glycemic index through resistant starch content increase [1]. *Moringa oleifera* leaf powder [27] added to wheat fresh pasta produced an increase in cooking loss and a reduced firmness with nutritional value enhancement given by higher phenol and mineral contents. The addition of coconut byproducts to wheat pasta led to lower firmness and color changes, while the fiber, protein, and lipid contents increased. Other studies [28] suggested the possibility of increasing durum wheat pasta fiber content by incorporating different vegetable powders (beet, carrot, and kale), along with significant changes in color. Fortification of durum wheat pasta with onion skin resulted in improved dietary fiber, ash, and total phenol content, as well as antioxidant activity, while technological properties diminished [29]. Furthermore, the reuse of seeds of Moldavian dragonhead as a pasta ingredient resulted in higher nutritional value without negative effects on the technological and sensory characteristics of the pasta [30]. Another study underlined the opportunity to increase the total polyphenol contents and antioxidant capacity of pasta via supplementation with olive pomace. The authors found a decrease in rapidly digestible starch, an increase in slowly digestible starch and resistant starch, and an improvement in other technological properties [31].

The addition of fiber-rich ingredients can have significant effects on dough rheological properties and on the final product texture, microstructure, and color. The effects of grape pomace on the composite flour and final product properties are relative to the addition level, while some negative effects of GP on dough rheology caused by gluten dilution can be minimized by particle size reduction [12,24]. Food texture, volume, and color are strongly affected by high levels of grape pomace addition; for this reason, some authors [26] suggested that amounts up to 6% GP can be incorporated into egg pasta without significant negative effects on the sensory acceptance. Furthermore, polyphenols can slow starch gelatinization via the interaction through hydrogen bonds with amylose molecules [32]. Grape skin phenolics, such as phenolic acids, tannins, and flavonoids, could have reducing effects on starch digestibility due to their abilities to inhibit enzyme activity or through the formation of starch–polyphenol complexes with resistance to enzyme attacks [33]. This study was aimed at evaluating the chance to exploit dried entire GPF as a source of phenolic compounds and fiber in order to produce fortified fettuccine. The choice of this byproduct as a functional ingredient can improve the nutritional profile of a widely consumed foodstuff such as pasta increasing the daily intake of bioactive components and adding economic value to the winemaking production chain. A number of studies have revealed the chance to use fiber- and polyphenol-rich ingredients in pasta; however, to our knowledge, there are no studies revealing the effects of whole GPF on common wheat dough for pasta production. The knowledge of the interactions of GPF with other components from wheat is very important for the development of functional pasta.

## 2. Materials and Methods

### 2.1. Chemical and Samples

Trans-Resveratrol was obtained from ICN Biomedicals (South Chillicothe Road, Aurora, OH, USA), whereas catechin, epicatechin, oenin, quercetin, quercetin-3-glucoside, and rutin were purchased from Extrasynthese (Genay, France). All other compounds such as solvents of analytical grade, were provided by Sigma-Aldrich (St. Louis, MO, USA).

### 2.2. Raw Material and Pasta Preparation

Wine pomaces (grape harvest year 2019) of *Vitis vinifera* L. cv. Lambrusco (achieved after fermentation) and Fiano (without fermentation) were collected from a local commercial winery (Cantele Winery Guagnano, Apulia Region, Italy). Pomace samples were dried in an oven at 40 °C, for 48 h in the dark. The dried GP was milled by a laboratory sample mill (FOSS, Hillerød, Denmark) to obtain powder, and then passed through a 1 mm sieve.

Pasta (fortified and nonfortified samples) was produced by s local pasta factory (www.pastificiodelduca.com, accessed on 31 May 2023, Parabita, Apulia Region, Italy). It was bronze-drawn pasta “fettuccine” made with durum wheat (*Triticum durum* L.) and water to help the development of gluten and salt. The development of pasta formulations consists of combining two ingredients: wheat flour and water (30%). This base formulation was modified by replacing 4% of wheat flour with 4% of GPF. For the preparation of pasta, the granulometry of wheat flour and mixed flours was adjusted to diameter ≤ 1 mm; then, the solid ingredients were mixed, before immediately adding the liquid ingredients until a homogeneous consistency of the dough was achieved. The drying process was carried out at temperatures below 40 °C for 36 h.

### 2.3. Phenolic Compound and Isoprenoid Extraction from Pasta and Characterization by HPLC Analysis

Phenolic compounds (soluble or insoluble) were extracted from 1 g of raw and cooked pasta as described by [34]. Briefly, aliquots of each sample were extracted twice with 80% *v*/*v* ethanol. The combined supernatants (soluble phenolic fraction) were collected, evaporated, and hydrolyzed with 2 M NaOH for 4 h. The insoluble phenolic acids were extracted from pellets by hydrolysis with 2 M NaOH for 4 h. Samples were acidified to pH 2.0 with 12 M HCl and extracted twice with ethyl acetate. The upper phase was collected, evaporated, dissolved in 80% ethanol, and assayed by HPLC-DAD as reported by [35]. Isoprenoids (tocochromanols or carotenoids) were obtained from 1 g of raw and cooked pasta as described by [34]. The final extracts were analyzed by HPLC-DAD as reported by [36].

### 2.4. Total Phenols Content

The Folin–Ciocâlteu method was carried out to evaluate the total amount of polyphenols. The total phenols (soluble or bound) in pasta extracts were assessed by determining the absorbance at 760 nm [35]. Results were expressed as milligrams of gallic acid equivalents per gram of pasta (mg GAEs/g of pasta).

### 2.5. Trolox Equivalent Antioxidant Capacity (TEAC) Assay

The Trolox equivalent antioxidant capacity (TEAC) assay in extracts containing soluble or bound phenolic compounds was used to determine the total antioxidant capacity as the ABTS^•+^-scavenging capacity. The analysis was performed as previously described by [35]. Absorbance was determined at 734 nm, and values were expressed as µmol Trolox equivalents (TE)/g of GPF pasta.

### 2.6. Oxygen Radical Absorbance Capacity (ORAC) Assay

The ORAC procedure was carried out following the procedure established by [11]. The antioxidant capacity of extracts for soluble or bound phenolic compounds was assayed using 96-well plates and an Infinite 200 Pro plate reader (Tecan, Männedorf, Switzerland). The antioxidant Trolox was used to make a standard curve (1–6 μM), and final ORAC values were expressed as μmol Trolox equivalents (TE)/g of GP pasta.

### 2.7. Fatty Acid Extraction from Pasta and Characterization by GC-MS

Total lipids were extracted from 0.1 g from pasta with 4 mL of n-hexane while stirring (3000 rpm) overnight at 4 °C. Samples were centrifugated (6000× *g* 10 min) and the organic phase was vaporized by a nitrogen stream. Lipids were subjected to fatty acid derivatization and analyzed by GC-MS as previous reported in [36].

### 2.8. Fibers, Ash, and Nitrogen Substances of GPF Pasta

Analyses of fibers, ash, and nitrogenous substances of GPF pasta were commissioned to an external analysis laboratory, which applied the following methods: AOAC 985.29 1986; Report ISTISAN 1996/34 pag. 77; MI 02.310 rev01 2020.

### 2.9. Color Evaluation of GPF Pasta

Whole GPF pasta color was evaluated using a Minolta CR-410 chromatometer (Konica Minolta Camera Co., Ltd., Osaka, Japan). The CIELAB color space was used to determine the parameters L* from black (0) to white (100), a* (red (+a) to green (−a) color), and b* (yellow (+b) to blue (−b) color).

### 2.10. Cooking Quality

Water absorption and cooking loss were determined as described by [37]. Ten grams of pasta was cooked for 8 min in 200 mL of boiling water. After cooking, the samples were drained for 3 min and weighed; the water absorption index (WAI) of cooked pasta was evaluated using the follow equation:$$WAI=\frac{\mathrm{weight}\;of\;cooked\;pasta\left(g\right)-weight\;of\;uncooked\;pasta(g)}{weight\;of\;uncooked\;pasta(g)}\times 100.$$

Cooking loss (CL) was assessed according to the AACC method 66–50 (AACC, 2000); in an air oven at 105 °C ± 2 °C, the pasta cooking water was dried. The residue was weighed and expressed as a percentage of the starting material using the follow equation:$$CL=\frac{weight\;of\;dry\;residue\left(g\right)}{weight\;of\;uncooked\;pasta\left(g\right)}\times 100.$$

The swelling index (SI) was determined as reported by [38] with some modifications. Briefly, pasta samples boiled and weighted for WAI determination were placed in glass containers, dried at 105 °C for 16 h, cooled, and weighted again. The results were calculated as follows:$$SI=\frac{weight\;of\;cooked\;pasta\left(g\right)}{\begin{array}{c} \mathrm{weight}\;\mathrm{of}\;\mathrm{cooked}\;\mathrm{pasta}\;\mathrm{after}\;\mathrm{drying}\;\left(g\right) \end{array}}.$$

### 2.11. Volatile Compound Extraction from Pasta and Characterization by GC-MS

Headspace solid-phase microextraction (HS-SPME) was employed to analyze volatile compounds according to [39] with some modifications. Briefly, 5 g of ground pasta samples, cooked or uncooked, were inserted into a 20 mL vial hermetically closed with a screw cap and a silicone septum. A 50/30 DVB-CAR-PDMS fiber (Supelco, Bellofonte, PA, USA) was used to adsorb volatiles for 30 min at 40 °C. GC–MS analyses were performed as reported by [40].

To evaluate the effect of the pasta’s volatile compounds, the ratio between the single peak area and the total molecule area in a specific chromatogram, expressed as a percentage, was considered.

### 2.12. Statistical Analysis

Values represent the mean ± standard deviation of three independent replicates.

A parametric method (two-sample-*t*-test) was used to evaluate the statistical significance of the differences between the measured data by means of the SigmaStat software Version 3.1 (Jandel Corp., Erkrath, Germany). The principal component analysis of volatile compounds was carried out using the Statistica 6.0 software package.

## 3. Results and Discussion

GP, an important agricultural byproduct in the winery industry, has emerged as an important cheap and sustainable source of polyphenols, minerals, and fibers. Among its active compounds, polyphenols can prevent and control different pathologies. For the reasons mentioned above, in addition to being an ideal food for nutritional enrichment, pasta was chosen to be the “vehicle” of the nutrients that GPF can add to the final foodstuff. However, it is worth mentioning that, in the production of pasta, certain criteria must be taken into account, such as technological and cooking quality, low cost, easy preparation, pleasant sensory aspects, and nutritional value. Bronze-drawn pasta is the most popular choice for those who prefer traditional dry pasta. By bronze-drawn pasta, we mean the extrusion of the water and semolina mixture through a bronze die, i.e., a matrix perforated with different shapes, to obtain the pasta. Bronze-drawn pasta is essentially opaque in color and not bold, even in the colored shapes; it is rough and porous to the touch, a fundamental characteristic that allows it to hold various sauces and condiments in an optimal manner [37]. From a nutritional point of view, bronze-drawn pasta, necessitating a more “stressful” process than Teflon-drawn pasta, requires a better-quality semolina. This, combined with drying at lower temperatures, results in a better nutritional profile, starting with the protein. The protein content of a good bronze-drawn pasta is 12–13% per 100 g. In this way, the present innovation refers to innovative pasta formulations based on wheat flour supplemented with whole GPF of any kind.

### 3.1. Chemical Composition and Polyphenol Content of GPF

The chemical composition of Fiano cv.-derived GPF was as follows: nitrogen substances (9.60 ± 0.85 g/100 g DW), fibers (58.60 ± 1.70 g/100 g DW), and ash content (5.27 ± 0.32 g/100 g DW). The TP content was 4.97 ± 0.53 gGAE/100 g DW. The chemical composition of Lambrusco cv.-derived GPF was as follows: nitrogen substances (14.00 ± 1.05 g/100 g DW), fibers (53.00 ± 1.44 g/100 g DW), and ash content (23.34 ± 1.11 g/100 g DW). The TP content was 1.96 ± 0.02 gGAE/100 g DW. Fibers are the major component of grape pomace, as already stated by Antonić et al. [10].

### 3.2. Phenol, Tocochromanol, and Carotenoid Characterization

An HPLC analysis of soluble and bound phenols was performed (Table 1A,B). In the raw pasta fortified with both white and red GP, the content of soluble phenolic molecules increased significantly, while a limited increase in bound phenols was measured in pasta supplemented with red GP pomace only (Table 1A). As shown in Table 1B, the cooking process determined the loss of bound phenols and the doubling of soluble phenols, both in control and GPF-supplemented pasta samples. Both tocochromanol and carotenoid contents increased in uncooked pasta supplemented with either white or red GPF (Table 1A). The cooking process determined a decrease in tocochromanol and carotenoid values; although featuring lower levels than raw samples, cooked fortified pasta still contained a higher amount of these molecules compared to the control (Table 1B). Phenols, tocochromanols, and carotenoids have antibacterial, antitumor, antioxidant, and anti-inflammatory properties [17,19]. These results indicated that Fiano and Lambrusco GPF-supplemented pasta is a functionalized food product enriched with bioactive molecules and a new production chain for the reuse of winery industry byproducts.

### 3.3. Antioxidant Activity of GPF-Supplemented Pasta

In accordance with the polyphenol amount, the addition of pomace flour increased antioxidant activity in comparison with the control pasta.

The antioxidant power was evaluated in the phenol extracts (soluble and bound) of control pasta and pasta with pomace flour added, before and after cooking (Table 2).

In the raw control pasta, free phenols were scarce and almost undetectable, whereas, in the raw pasta with pomace flour, both white and red, they were definitely detectable. In the extracts of bound phenols, the TP, TEAC, and ORAC values were almost twice as high in the raw dough with added grape pomace flour as in the control.

In each pasta sample, cooking greatly lowered the antioxidant capability. Moreover, bound phenols disappeared in all cooked pasta samples, and their antioxidant capacity could not be measured. After cooking, the antioxidant activity of soluble phenol fractions (both TEAC and ORAC) remained about 20 times higher than that of the control in pasta samples containing white pomace flour, and more than 10 times (from 11- to 13-fold) in pasta samples containing red pomace flour. The difference in antioxidant activity between the pasta sample obtained with white and red GP was related to the different phenolic compositions. Thus, it can be hypothesized that the different phenolic composition of the two pasta samples W and R led to a different response after cooking due to the presence of anthocyanins, which in the red, underwent degradation.

### 3.4. Chemical Properties of GPF-Supplemented Pasta

The incorporation of GPF results in a compact microstructure of pasta [1,12]. Proteins, fibers, and polyphenols of GPF connect with gluten proteins to reinforce the dough protein matrix. The presence of GP fibers modifies the absorbance of water in the mixed flour and consequently increase the roughness of the dough surface [12].

The presence of whole GPF increased the fiber component before and after cooking (Table 3). Furthermore, the bioavailability of polyphenols was directly related to interactions with fiber. Supplementation of fiber-rich ingredients is a smart technique to enhance the nutritional value of the final product [1]. Fiber is known to have a number of biological functions, such as activity against cancer, improving the functioning of gastrointestinal systems, improving the activity of the cardiovascular system, and decreasing cholesterol and blood glucose levels [13].

The ash of food is the solid residue resulting from the complete combustion of organic matter. During combustion, sugars, starches, fibers, proteins, organic acids, vitamins, etc. are burnt, sparing the minerals, which accumulate in the ash in the form of oxides, carbonates, sulfates, and phosphates. Food ashes, if rich in oxides of potassium, magnesium, calcium, iron, etc., are defined as alkaline, as they can form alkaline aqueous solutions. The ash content reflects the mineral content shown in a previous paper [41]. The ideal diet should have a slightly alkaline balance, which can be obtained from a diet rich in fruit and vegetables, with moderate amounts of animal protein, as most food science professionals suggest. In the case of pasta samples, the ash residue showed a slight increase after the addition of pomace flour in raw pasta; however, after cooking, ash content decreased in both control and GPF-supplemented pasta (Table 3). As described in Table 3, the nitrogen substances contained in the pasta samples appeared to be only slightly increased in the uncooked pasta with red grape pomace added; especially after cooking, the nitrogenous substance content did not seem to change. This result confirms that cooking pasta had no significant effect on protein pasta content as reported by Tazrart et al. [42].

There was a significant increase in fiber content in the uncooked W and R pasta samples, as well as in the R samples after cooking, compared to cooked and uncooked control pasta samples. Cooking affects the fiber content of pasta, causing a slight loss of soluble dietary fibers [43]; however, but the main part of dietary fibers in GPF comprises insoluble fibers [44], resulting in minor fiber loss after cooking of fortified pasta. Non-amidic polysaccharides from vegetables are beneficial polymers of cellulose, xyloglucan, arabinan, pectin, lignin, and structural proteins. They are recognized as dietary fiber, and dietary fiber intake of 25–30 g/day is associated with health gains [22]. Polysaccharides associated with polyphenols are recognized as potential dietary fibers with antioxidant properties [9]. Grape pomace reuse as a secondary raw material in new food production can combine antioxidant properties with fiber benefits. Taking into account the low-cost processing and integral use of the pomace described in this study, pasta and other foodstuffs with higher bio-functional quality and sustainability should be produced.

The cooking time of our pasta (8–9 min) is not experimental. The data in Table 4 indicate the technological parameters linked to the cooking of pasta samples, such as the cooking loss, swelling, and water absorption indices. These parameters are important in assessing pasta quality by both consumers and industry [16]. In particular, the cooking loss measures the release of starch and soluble elements from the pasta surface into the cooking water; low CL values are related to good-quality pasta [45]. The data in Table 4 show a slight increase in CL, SI, and WAI in pasta supplemented with GPF when compared to the control pasta value. The increase in cooking parameters was greater for pasta supplemented with red grape-derived pomace. The cooking parameter results in Table 4 reflect the higher fiber content of cooked pasta R samples (Table 3) because of the capacity of fibers to absorb water and to interfere with the semolina protein–starch matrix. The GPF-supplemented pasta can be considered good-quality because the CL was <12%, as indicated by Sant’Anna et al. [16], and even <8% (CL value of semolina spaghetti) [46].

Our starting material was total pomace flour (peel, seeds, and stalks) which involved lower work production and energy consumption with zero waste. The presence of the grape seeds enriched the flour with fatty acids and fat-soluble vitamins (vitamins A and E). Checking the shelf-life of the flour, we showed that the product was stable for 6 months at 4 °C [36]. Table 5 reports the fatty acid percentage and distribution (SFA, MUFA, and PUFA) in uncooked and cooked pasta. The results evidence that, before cooking, in the W sample, the MUFA percentage was significantly higher than in C; on the other hand, in W and R cooked pasta, the PUFA percentages were higher than in C.

The addition of 4% GPF derived from both Fiano and Lambrusco cv. significantly modified the pasta color. As shown in Table 3, luminosity (L*) and yellowness (b*) decreased in cooked and uncooked pasta supplemented with both white grape- and red grape-derived pomace flour. An increase in redness and a decrease in green tone (a*) were also evident after the addition of grape pomace in both uncooked and cooked fettuccine, to a major extent for the latter. The decrease in L* values of GPF-fortified pasta indicates a darkening of the samples due to the incorporation of a blackish ingredient into the pasta dough, as reported by [23]. An increase in a* values is related to an ingredient rich in anthocyanins and tannins. The cooking process resulted in a darker fortified pasta, which is compatible with the findings of the literature [1,16,38].

### 3.5. Volatiles Compounds in GPF-Supplemented Pasta

SPME-GC/MS was applied to characterize the headspace volatile fraction of paste samples, enriched with pomace from two Italian varieties of Lambrusco (R) and Fiano (W), before and after cooking (1 and 2). As depicted in Table 6, 32 molecules were identified, such as esters, alcohols, aldehydes, ketones, terpenes, and furan compounds. In uncooked samples, R featured the most numerous and the most abundant esters (56.51%) and alcohols (19.33%) compared with the control and W samples, suggesting a more pronounced flavor than other pasta samples.

The significative ester percentage in cooked R was probably contributed by the pomace due to the typical fermentative activities of winemaking [47,48]. High aldehyde contents were found in the control and uncooked R samples, with percentages varying between 33.29 and 29.94, respectively. As reported in Table 6, an increase in aldehydes and ketones could be found in cooked R and W pasta, along with a decrease in esters and alcohols in all samples, likely due to the drying process, which particularly led to a decrease in the esters and alcohols. Indeed, during the cooking process, alcohol loss is promoted by the solubility of this class of molecules in water [49].

Among the aldehydes and ketones, hexanal was found to be the most abundant compound in the volatile fraction of the pasta, as reported by others [35,40], with a percentage ranging from 2.04 in the Fiano pasta to 4.32 in the Lambrusco sample. Hexanal, heptanal, octanal, and nonanal, are the products of oxidative degradation of unsaturated fatty acids (oleic and linoleic acid) [50]. The increase in aldehydes, during the cooking process, was probable due to the release of starch and lipids that become available through thermal oxidation enhanced by high temperatures [50,51]. Fatty acid oxidation was also responsible for the production of other key volatile molecules such as 1-octene-3-one and 2-pentyl furan. The content of the latter increased significantly after cooking, three times in W and two times in cooked R and C. Notably, aldehydes and 2-pentyl furan are also considered key molecules of pasta aroma, as reported by [49]. Indeed, they have a low odor threshold and reduced contribution to pasta flavor [52]. Among the aldehydes, another important molecule is benzaldehyde, a product of phenylalanine degradation [53], detected in higher percentages in cooked samples, ranging from 2.58 in W (Fiano cv.) to 6.91 in R (Lambrusco cv.) samples.

Principal component analysis (PCA) was used as an exploratory statistical technique to investigate which variables play a major role in discriminating between cooked and uncooked samples. Figure 1 shows the sample (score) and variable (loading) plots corresponding to the first two eigenvectors, which described 77.3% of the total variance (PC1: 42.0%; PC2: 35.4%). PC1 clearly discriminated the cooked pasta samples, R2 and W2, from both the respective uncooked samples and the two controls. The loading plot showed that this is linked to the presence of the higher number of identified volatile compounds, including aldehydes, alcohols, and esters. The addition of GPF in combination with the cooking step had a relevant effect on the volatile profile of R2. The same type of effect, albeit less pronounced, was also detectable for sample W2.

The control, on the other hand, both before and after cooking, appeared to be poorly correlated with the volatile molecules analyzed, suggesting a neutral sensory profile. PC2, on the other hand, highlighted the significant differences between the two samples fortified with red pomace, R1, and R2. R1, with a high PC2 (score 6.47) and negative PC1 (−2.18), correlated positively with esters; with cooking, esters decreased and aldehydes increased, characterizing sample R2, which clustered between positive PC1 and PC2.

Multivariate analysis confirmed that most VOCs were associated with pasta enriched with aromatic Lambrusco grape pomace, followed by Fiano pomace.

## 4. Conclusions

The results of this study indicate that is possible to obtain pasta with increased antioxidant activity, increased polyphenol content, increased tocochromanols and carotenoids, increased fiber content, and not modified or slightly increased fatty acid content, following the inclusion of grape pomace flour in the dough formulation. Moreover, pomace, red and/or white, can be successfully used as an additive in pasta samples, improving the volatile profile with increased esters, terpenes, alcohols, and aldehydes compared to the control. To our knowledge, this is the first paper on the production of bronze-drawn pasta made with controlled percentages of flour obtained from whole pomace flour including grape skins, grape seeds, and stems. In this study, we highlighted the integration of different approaches for valorizing the GP (*Vitis vinifera* L.) in order to improve a zero-waste pathway for this wine-chain byproduct. Following this approach, GP can be considered a high-value resource to be converted into sustainable food and bio-functional ingredients, while fully respecting the circular economy concept.

## Figures and Tables

**Figure 1 foods-12-02593-f001:**
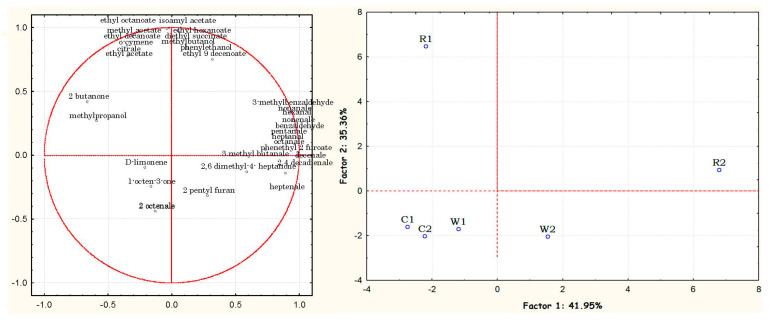
Principal component analysis applied to dataset of samples and 32 volatile compounds. C = control; W = white grape pomace flour from Fiano cv.; R = red grape pomace flour from Lambrusco cv.; 1 = uncooked; 2 = cooked.

**Table 1 foods-12-02593-t001:** Composition of phenols (soluble and bound), tocochromanols, and carotenoids in uncooked and cooked control and GPF-supplemented pasta.

**(A)**						
**Uncooked**						
	**C**		**W**		**R**	
			**μg/g FW**			
**Phenols**	**Soluble**	**Bound**	**Soluble**	**Bound**	**Soluble**	**Bound**
Gallic acid	nd	nd	3.33 ± 0.06	nd	1.91 ± 0.01	nd
Coutaric acid	nd	0.05 ± 0.001	nd	nd	nd	nd
Catechin	nd	nd	3.89 ± 0.05	nd	nd	nd
Epicatechin	nd	5.18 ± 0.02	1.53 ± 0.04	4.36 ± 0.04	4.67 ± 0.04	3.33 ± 0.03
Vanillic acid	nd	7.72 ± 0.03	nd	nd	nd	11.41 ± 0.19
Sinapic acid	nd	10.72 ± 0.39	2.75 ± 0.02	9.25 ± 0.52	nd	7.85 ± 0.05
Syringic acid	nd	nd	nd	nd	nd	18.65 ± 0.75
*p*-Coumaric acid	1.31 ± 0.01	4.76 ± 0.06	nd	6.63 ± 0.05 *	1.44 ± 0.01 *	34.47 ± 0.40 *
Total anthocyanins	nd	nd	0.26 ± 0.001	0.52 ± 0.03	2.08 ± 0.02	0.64 ± 0.02
Quercetin	nd	nd	1.72 ± 0.02	nd	nd	19.14 ± 0.17
Rutin	nd	13.48 ± 0.98	nd	5.57 ± 0.02 *	nd	4.03 ± 0.05 *
Quercetin-3-Glc	nd	0.46 ± 0.001	0.48 ± 0.01	0.86 ± 0.01	nd	nd
Oenin	nd	nd	nd	nd	nd	nd
Kaempferol	nd	nd	0.71 ± 0.02	0.60 ± 0.001	nd	nd
Kaempferol-3-Glc	nd	nd	0.37 ± 0.02	0.24 ± 0.09	0.35 ± 0.01	0.15 ± 0.01
Kaempferol-3-Rut	nd	nd	nd	0.14 ± 0.01	nd	nd
Ferulic acid	0.14 ± 0.02	224.02 ± 2.71	1.03 ± 0.01 *	210.80 ± 1.14 *	0.11 ± 0.02	186.41 ± 0.71 *
Caftaric acid	nd	nd	nd	nd	nd	nd
***Total***	** *1.45 ± 0.03* **	** *266.39 ± 4.19* **	** *16.07 ± 0.25 ** **	** *238.67 ± 1.91 ** **	** *10.57 ± 0.11 ** **	** *286.08 ± 2.38 ** **
**Tocochromanols**						
β-tocotrienols	1.81 ± 0.03		5.29 ± 0.05 *		3.65 ± 0.02 *	
α-tocopherols	nd		nd		0.89 ± 0.03	
***Total***	** *1.81 ± 0.03* **		** *5.29 ± 0.05 ** **		** *4.54 ± 0.05 ** **	
**Carotenoids**						
Lutein	0.72 ± 0.01		1.79 ± 0.01 *		1.16 ± 0.05 *	
Zeaxanthin	0.01 ± 0.0005		0.03 ± 0.0005 *		0.02 ± 0.002 *	
β-carotene	nd		0.04 ± 0.001		0.05 ± 0.001	
***Total***	** *0.73 ± 0.03* **		** *1.86 ± 0.01 ** **		** *1.22 ± 0.05 ** **	
**(B)**						
**Cooked**						
	**C**		**W**		**R**	
			**μg/g FW**			
**Phenols**	**Soluble**	**Bound**	**Soluble**	**Bound**	**Soluble**	**Bound**
Gallic acid	nd	nd	1.88 ± 0.01	nd	1.01 ± 0.02	nd
Coutaric acid	nd	nd	nd	nd	nd	nd
Catechin	nd	nd	4.66 ± 0.09	nd	1.81 ± 0.08	nd
Epicatechin	nd	nd	13.26 ± 0.99	nd	nd	nd
Vanillic acid	nd	nd	2.37 ± 0.01	nd	2.77 ± 0.04	nd
Sinapic acid	1.19 ± 0.01	nd	1.76 ± 0.003 *	nd	nd	nd
Syringic acid	nd	nd	nd	nd	1.01 ± 0.16	nd
*p*-Coumaric acid	0.54 ± 0.01	nd	0.98 ± 0.03 *	nd	0.88 ± 0.01 *	nd
Total anthocyanins	nd	nd	nd	nd	2.23 ± 0.02	nd
Quercetin	nd	nd	2.13 ± 0.08	nd	7.73 ± 0.18	nd
Rutin	0.03 ± 0.001	nd	8.44 ± 0.07 *	nd		nd
Quercetin-3-Glc	nd	nd	0.44 ± 0.01	nd	1.50 ± 0.007	nd
Oenin	nd	nd	nd	nd	0.53 ± 0.002	nd
Kaempferol	nd	nd	0.49 ± 0.02	nd	0.26 ± 0.001	nd
Kaempferol-3-Glc	nd	nd	0.48 ± 0.03	nd	nd	nd
Kaempferol-3-Rut	nd	nd	nd	nd	nd	nd
Ferulic acid	0.64 ± 0.01	nd	1.93 ± 0.08 *	nd	0.91 ± 0.01 *	nd
Caftaric acid	nd	nd	nd	nd	nd	nd
***Total***	** *2.40 ± 0.03* **		** *39.26 ± 1.42 ** **		** *20.64 ± 0.53 ** **	
**Tocochromanols**						
β-tocotrienols	1.35 ± 0.01		2.50 ± 0.13 *		3.43 ± 0.09 *	
α-tocopherols	nd		nd		0.46 ± 0.01	
***Total***	** *1.35 ± 0.01* **		** *2.50 ± 0.13 ** **		** *4.19 ± 0.10 ** **	
**Carotenoids**						
Lutein	0.61 ± 0.01		0.92 ± 0.06 *		0.99 ± 0.01 *	
Zeaxanthin	0.01 ± 0.0004		0.03 ± 0.002 *		0.03 ± 0.003 *	
β-carotene	nd		0.03 ± 0.0004		0.04 ± 0.003	
***Total***	** *0.62 ± 0.01* **		** *0.99 ± 0.06 ** **		** *1.06 ± 0.02 ** **	

C = control pasta; W = pasta supplemented with white grape pomace (Fiano cv.) flour; R = pasta supplemented with red grape (Lambrusco cv.) pomace flour. * Statistically significant differences (*p* < 0.05) between each sample and the control pasta as determined by the two-sample *t*-test.

**Table 2 foods-12-02593-t002:** Antioxidant activity of polyphenol extracts from uncooked and cooked control and GPF-supplemented pasta.

**Uncooked**	**C**		**W**		**R**	
	**Soluble P.**	**Bound P.**	**Soluble P.**	**Bound P.**	**Soluble P.**	**Bound P.**
	*μmolTE/g*					
**TEAC**	0.104 ± 0.02	3.52 ± 0.15	1.21 ± 0.04 *	7.56 ± 0.32 *	1.02 ± 0.02 *	6.8 ± 0.38 *
	*μmolTE/g*					
**ORAC**	0.3 ± 0.0	5.8 ± 0.004	2.03 ± 0.26 *	10.47 ± 2.45 *	2.19 ± 0.0 *	9.48 ± 0.4 *
	*μg GAEs/g*					
**TP**	nd	225.37 ± 26.0	68.7 ± 2.6 *	441.44 ± 3.44 *	66.333 ± 3.77 *	398.136 ± 3.82 *
**Cooked**	**C**		**W**		**R**	
	**Soluble P.**	**Bound P.**	**Soluble P.**	**Bound P.**	**Soluble P.**	**Bound P.**
	*μmolTE/g*					
**TEAC**	0.11 ± 0.016	nd	2.36 ± 0.06 *	nd	1.43 ± 0.02 *	nd
	*μmolTE/g*					
**ORAC**	0.25 ± 0.046	nd	4.9 ± 0.28 *	nd	2.88 ± 0.001 *	nd
	*μg GAEs/g*					
**TP**	nd	nd	153.0 ± 0.86 *	nd	101.314 ± 1.26 *	nd

C = control pasta; W = pasta supplemented with white grape pomace (Fiano cv.) flour; R = pasta supplemented with red grape (Lambrusco cv.) pomace flour; TP = total phenols; P = phenols. * Statistically significant differences (*p* < 0.05) between each sample and the control pasta, as determined by the two-sample-*t*-test.

**Table 3 foods-12-02593-t003:** Chemical composition and color analysis of uncooked and cooked control pasta and GPF-supplemented pasta.

Sample	Ash	Fibers	Nitrogen Substances	L*	a*	b*
Uncooked		**%DW**				
C	0.70 ± 0.03	4.1 ± 0.7	13.88 ± 1.72	77.27 ± 0.47	1.15 ± 0.09	27.51 ± 0.51
W	0.78 ± 0.03 *	5.3 ± 0.5 *	14.9 ± 1.91	54.98 ± 0.22 *	4.55 ± 0.05 *	9.47 ± 0.10 *
R	0.78 ± 0.03 *	6.7 ± 0.6 *	15.36 ± 1.97 *	50.31 ± 0.19 *	2.85 ± 0.04 *	3.38 ± 0.02 *
Cooked						
C	0.55 ± 0.02	2.5 ± 0.4	15.07 ± 1.93	77.79 ± 0.38	−2.01 ± 0.15	37.72 ± 0.35
W	0.49 ± 0.05	3.3 ± 0.5	15.0 ± 1.92	40.44 ± 0.16 *	9.02 ± 0.03 *	13.99 ± 0.12 *
R	0.49 ± 0.5	6.2 ± 0.6 *	15.53 ± 1.99	35.52 ± 0.28 *	5.86 ± 0.01 *	4.18 ± 0.02 *

C = control pasta; W = pasta supplemented with white grape pomace (Fiano cv.) flour; R = pasta supplemented with red grape (Lambrusco cv.) pomace flour; DW = dry weight. * Statistically significant differences (*p* < 0.05) between each sample and the control pasta as determined by the two-sample *t*-test.

**Table 4 foods-12-02593-t004:** Technological properties of cooked control pasta and pasta fortified with grape pomace.

Sample	Cooking Loss (%)	Swelling Index (g/g DW)	Water Absorption Index (%)
C	3.88 ± 0.09	2.48 ± 0.07	119.37 ± 5.00
W	4.61 ± 0.11 *	2.64 ± 0.05 *	127.83 ± 3.79
R	5.03 ± 0.09 *	2.75 ± 0.04 *	135.73 ± 3.99 *

C = control pasta; W = pasta fortified with white grape pomace flour; R = pasta fortified with red grape pomace flour; DW = dry weight. * Statistically significant differences (*p* < 0.05) between each sample and the control pasta as determined by the two-sample *t*-test.

**Table 5 foods-12-02593-t005:** Comparison of fatty acid content of control pasta and pasta supplemented with GPF.

	SFA	MUFA	PUFA
Uncooked			
C	22.25 ± 1.48	23.18 ± 2.99	54.57 ± 4.31
W	22.24 ± 1.86	24.68 ± 1.52 *	53.08 ± 4.47
R	22.18 ± 1.85	23.18 ± 2.99	54.22 ± 4.84
Cooked			
C	33.66 ± 2.24	19.87 ± 2.35	46.47 ± 4.03
W	25.63 ± 1.33 *	21.14 ± 1.2	53.23 ± 3.87 *
R	27.84 ± 1.77 *	20.87 ± 1.52	51.29 ± 2.02 *

C = control pasta; W = Fiano cv.; R = Lambrusco cv.; SFA = saturated fatty acid; MUFA = monounsaturated fatty acids; PUFA = polyunsaturated fatty acid. * Statistically significant differences (*p* < 0.05) between each sample and control pasta as determined by the two-sample *t*-test.

**Table 6 foods-12-02593-t006:** Volatile compounds of pasta samples cooked and uncooked. Data are expressed as the percentage of the total GC area.

	C	W	R	C	W	R	Tukey Test
	Uncooked Pasta			Cooked Pasta			
	AREA% ± SD						
**Esters**							
Methyl acetate	-	0.15 ± 0.04	2.04 ± 0.12	-	-	-	*
Ethyl acetate	10.26 ± 2.71	-	16.53 ± 4.20	6.24 ± 1.34	-	5.60 ± 0.62	*
Isoamyl acetate	-	0.21 ± 0.04	1.87 ± 0.21	-	0.11 ± 0.03	0.54 ± 0.08	*
Ethyl hexanoate	-	0.44 ± 0.07	6.03 ± 1.52	-	0.18 ± 0.04	1.85 ± 0.34	*
Ethyl octanoate	3.74 ± 0.91	0.42 ± 0.07	15.71 ± 4.11	1.87 ± 0.42	0.15 ± 0.04	2.28 ± 0.34	*
Ethyl decanoate	2.79 ± 0.63	0.89 ± 0.21	12.49 ± 4.27	1.35 ± 0.42	0.54 ± 0.15	2.06 ± 0.34	*
Diethyl succinate	-	-	0.65 ± 0.06	-	-	0.34 ± 0.04	*
Ethyl 9 decanoate	-	0.87 ± 0.33	1.19 ± 0.08	-	0.25 ± 0.05	0.93 ± 024	*
Phenethyl 2 furoate	-	-	-	-	0.58 ± 0.12	3.25 ± 0.64	*
** *Total* **	** *16.79* **	** *2.98* **	** *56.51* **	** *9.46* **	** *1.81* **	** *16.84* **	
**Aldehydes and ketones**							
2-Butanone	15.95 ± 3.83	0.16 ± 0.05	14.08 ± 4.10	6.54 ± 1.27	-	-	*
3-Methyl butanal	-	0.72 ± 0.15	0.48 ± 0.07	-	1.21 ± 0.07	1.01 ± 0.02	*
Pentanal	-	-	-	-	-	0.95	
Hexanal	-	2.04 ± 0.24	4.32 ± 0.18	-	6.79 ± 1.22	13.60 ± 2.07	*
2.6-Dimethyl-4 heptanone	13.80 ± 4.66	0.30 ± 0.04	7.94 ± 2.10	7.81 ± 1.24	23.98 ± 5.17	18.85 ± 3.41	*
Heptanal	-	-	-	-	-	1.69 ± 0.22	
Octanal	-	-	-	-	-	2.14 ± 0.08	
2-Heptenale	-	2.51 ± 0.16	0.25 ± 0.07	-	2.60 ± 0.81	4.11 ± 0.83	*
Nonanal	-	0.06 ± 0.02	0.54 ± 0.08	-	0.75 ± 0.21	2.05 ± 0.14	*
1-Octen-3-one	-	0.10 ± 0.02	-	-	-	-	
2-Octenale	3.54 ± 0.25	4.25 ± 0.61	-	-	0.85 ± 0.18	1.35 ± 0.18	*
Benzaldehyde	-	1.30 ± 0.14	1.14 ± 0.04	-	2.58 ± 0.51	6.91 ± 2.31	*
2-Nonenale	-	0.12 ± 0.03	0.35 ± 0.11	-	0.69 ± 0.15	1.88 ± 0.42	*
3-Methylbenzaldehyde	-	0.52 ± 0.15	0.85 ± 0.20	-	1.02 ± 0.24	2.18 ± 0.34	*
2-Decenale	-	-	-	-	0.60 ± 0.11	1.55 ± 0.45	*
2.4-Decadienale	-	-	-	-	1.59 ± 0.34	3.35 ± 0.73	
** *Total* **	** *33.29* **	** *12.07* **	** *29.94* **	** *14.35* **	** *42.65* **	** *61.62* **	
**Terpenes**							
d-Limonene	-	2.16 ± 0.12	0.35 ± 0.07	-	-	-	*
o-Cymene	-	-	0.24 ± 0.05	-	-	-	
Citrale	-	0.21 ± 0.04	0.60 ± 0.14	-	-	-	*
** *Total* **		** *2.37* **	** *1.19* **				
**Alcohols**							
2-Methyl-propanol	13.47 ± 2.34	-	10.02 ± 2.04	9.64 ± 1.47	-	2.58 ± 0.81	*
3-Methyl-1-butanol	-	0.92 ± 0.21	3.57 ± 0.74	-	0.14 ± 0.04	1.47 ± 0.15	*
Phenylethanol	-	0.58 ± 0.11	5.74 ± 0.88	-	0.15 ± 0.03	3.54 ± 0.72	*
** *Total* **	** *13.47* **	** *1.50* **	** *19.33* **	** *9.64* **	** *0.29* **	** *7.59* **	
**Furanic compounds**							
2 Pentyl furan	6.14 ± 1.05	2.48 ± 0.62	4.27 ± 0.65	11.25 ± 2.18	7.63 ± 1.84	8.61 ± 1.92	*

C = control; W = white grape pomace flour from Fiano cv.; R = red grape pomace flour from Lambrusco cv. * *p* ≤ 0.05 according to the Tukey test; SD = standard deviation; -not detected.

## Data Availability

All authors of the article are available to share the research data upon request.

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
