# Peer review of "Whole Grape Pomace Flour as Nutritive Ingredient for Enriched Durum Wheat Pasta with Bioactive Potential"

_foods, 2023, doi:10.3390/foods12132593_

Round 1
Reviewer 1 Report
Manuscript recieved for review investigates the effect of whole grape pomace flour introduction to the durum wheat pasta formulation on different nutritive and quality characteristics.
Intorduction section needs some corrections regarding cited literature.
All detailed comments are noted in pdf file.
Material and methodes section is appropriate for the described and conducted testing. Applied testing of nutritive compaunds with bioactive potential are very comperhensive.
Results and discussion section is very elaborate. Quality of results’ discussion is very good, with very elaborate referencting to other authors.
Conclusion section needs some corrections.
Decission: minor revision

Author Response
Whole grape pomace flour as nutritive ingredient for enriched durum wheat pasta with bioactive potential
Manuscript ID : foods-2456751
Responses to reviewer 1
The authors appreciate comments of Reviewer 1.
Reviewer will find all modified parts highlighted in yellow in the new manuscript version.
Introduction section needs some corrections regarding cited literature.
Authors added references in Introduction, lines 29, 35, 42 and 46
Merge table 3 to 6 in one table.
Authors merge table 3 and 6:
|
Sample |
|
Ash |
|
Fibres |
|
Nitrogen substances |
|
L* |
a* |
b* |
|
|
|
|
|
%DW |
|
|
|
|
|
|
|
C1 |
|
0.70±0.03 |
|
4.1±0.7 |
|
13.88±1.72 |
|
77.27±0.47 |
1.15±0.09 |
27.51±0.51 |
|
W1 |
|
0.78±0.03* |
|
5.3±0.5* |
|
14.9±1.91 |
|
54.98±0.22* |
4.55±0.05* |
9.47±0.10* |
|
R1 |
|
0.78±0.03* |
|
6.7±0.6* |
|
15.36±1.97* |
|
50.31±0.19* |
2.85±0.04* |
3.38±0.02* |
|
C 2 |
|
0.55±0.02 |
|
2.5±0.4 |
|
15.07±1.93 |
|
77.79±0.38 |
-2.01±0.15 |
37.72±0.35 |
|
W2 |
|
0.49±0.05 |
|
3.3±0.5 |
|
15.0±1.92 |
|
40.44±0.16* |
9.02±0.03* |
13.99±0.12* |
|
R2 |
|
0.49±0.5 |
|
6.2±0.6* |
|
15.53±1.99 |
|
35.52±0.28* |
5.86±0.01* |
4.18±0.02* |
This statement is more suitable for Conclusion section. Either rephase or move to conclusion
The Authors thank for comment. Lines 538-540 were removed, and the phrase moved to conclusion (lines 547-549).
Figure 1:
Fig 1 title moved below the figure and “,” has been replaced with “.”.
About Reviewer suggestion to merge the plots authors think that to make the distribution of the molecules with respect to the samples on the plane clearer, to avoid the superimposition of molecules and sample names, and to avoid merging plots with such different axis extensions, would be better to separate the two plots.

Reviewer 2 Report
The amount of grape pomace flour in the pasta formulation must be precisely determined - compare lines 133 and 247 (Was it addition or flour substitution?).
Section 2.3: Please provide the principle of extraction of phenolic compounds (What solvent, what conditions?) Replace "and" by "or" in lines 142, 144 and 213.
Line 157 and elsewhere: reword "phenolic extracts (soluble and bound)"
Section 2.10: Be consequent in providing formulas for all the parameters (indices) determined.
Section 3, lines 246-250: Move these information to the Materials and Methods.
Pay attention to the captionevs of the Tables - explanations of symbols should be included in the footer of the Tables. Avoid duplication of information (eg. Control pasta).
Table 1. Reconsider differences between data referring to Epicatechin and Sinapic acid.
Table 2. Caption. There is no AA symbol used in the Table.
Table 3. Does really data in Table 3 present any physical properties of the samples? Reconsider the Table caption. Explain if the percentage values are calculated as dry mass basis or not. What nitrogen compounds are included in the term "Nitrogen substances"? The changes in the fibre and nitrogen substances contents resulting from the GP addition or cooking are not explained. For example, why fibre content of C2 is lower that that of C1?
Lines 416-417: What is "positive limits of pasta quality"?
Table 4. The results are poorly discussed. Explain abbreviation for DW.
Lines 439-441: Explain the probable reasons for such results. Provide expamples of MUFA and PUFA.
Lines 502-503: What are "unripened fatty acids"?
Table 7: Should be "Tukey test".
Moderate editing of English language required (including punctuation).
Author Response
Whole grape pomace flour as nutritive ingredient for enriched durum wheat pasta with bioactive potential
Manuscript ID : foods-2456751
Responses to reviewer 2
The authors thanks Reviewers 2 for comments and considerations.
Reviewer will find all modified parts highlighted in green in the new manuscript version.
The amount of grape pomace flour in the pasta formulation must be precisely determined - compare lines 133 and 247 (Was it addition or flour substitution?).
Pasta formulation has been revised. 4% of semolina has been substituted with grape pomace flour (lines 133-137).
Section 2.3: Please provide the principle of extraction of phenolic compounds (What solvent, what conditions?) Replace "and" by "or" in lines 142, 144 and 213.
Thanks for your comments. Phenolic compounds extraction has been described in detail in lines 143-148. Suggested replacements of “and” by “or”, were done.
Line 157 and elsewhere: reword "phenolic extracts (soluble and bound)"
The revision of the sentence has been carried out: see lines 162-163, 171.
Section 2.10: Be consequent in providing formulas for all the parameters (indices) determined.
Formulas for cooking loss and swelling index have been added to revised manuscript.
Section 3, lines 246-250: Move these information to the Materials and Methods.
Thanks for your comment. The sentence had been moved to M&M (lines 133-137).
Pay attention to the captions of the Tables - explanations of symbols should be included in the footer of the Tables. Avoid duplication of information (eg. Control pasta).
Captions of table 1, 2, 3, 4, 5 and 6 had been modified following reviewer’s suggestions.
Table 1. Reconsider differences between data referring to Epicatechin and Sinapic acid.
Thanks for your comment. It is known from literature the impact of most common cooking method on physico-chemical and nutritional properties of foods. The impact on phytochemical content is variable and related to type of food products, plant ingredient genotype, growing location and condition, type of matrix in which the compounds are included, food processing techniques and cooking methods. Particularly, cooking by boiling dry pasta have impact on its phenolic profile. Literature reports losses of some phenolic classes in cooking water and a reduction of bound phenol fraction. Our results confirm this trend, showing a decrease of bound phenolic compounds and an increase of soluble phenolic compounds of cooked pasta. Bound epicatechin and sinapic acid were not detectable after cooking in every sample while soluble epicatechin was quantified in pasta added with GPF derived from Fiano grape. Soluble sinapic acid was detectable in cooked control pasta and in pasta added with GPF derived from Fiano grape.
Table 2. Caption. There is no AA symbol used in the Table.
AA symbol has been deleted from Table 2 caption.
Table 3. Does really data in Table 3 present any physical properties of the samples? Reconsider the Table caption.
Table 3 Caption has been revised following Reviewer suggestions.
Explain if the percentage values are calculated as dry mass basis or not.
Yes, percentage was calculated as dry weight basis and it was described in table 3 revised version.
What nitrogen compounds are included in the term "Nitrogen substances"?
In this assay method, nitrogen content has been converted in protein content by a nitrogen-to-protein conversion factor set at 6.25.
The changes in the fibre and nitrogen substances contents resulting from the GP addition or cooking are not explained.
Thanks for your comment. The authors have included in the text a more in-depth discussion of the results in Table 3 (lines 386-397).
Lines 416-417: What is "positive limits of pasta quality"?
Table 4. The results are poorly discussed. Explain abbreviation for DW.
Thanks for your comments. The authors have included in the text a more in-depth discussion of the results in Table 4 (lines 415-426)
Lines 439-441: Explain the probable reasons for such results. Provide expamples of MUFA and PUFA.
In the W sample MUFA percentage was significantly higher than C for the presence of palmitoleic acid and the highest content of oleic acid (data not shown).
The presence of antioxidants in W an R cooked, such as polyphenols, tocochromanols and carotenoids, could play an important role in oxidative protection to the PUFA in enriched pasta.
We add the table reporting the single fatty acids compounds identified in the samples:
|
|
C1 |
|
W1 |
|
R1 |
|
Miristic acid |
0.25±0.01 18.38±1.21 3.21±0.23 0.30±0.001 0.36±0.03 nd 20.15±1.87 1.36±0.23 1.31±0.87 0.36±0.02 49.49±3.45 4.81±0.82 0.27±0.04 22.25±1.48 23.18±2.99 54.57±4.31 |
|
0.16±0.01 12.98±1.12 7.79±0.63 0.77±0.06 0.54±0.04 0.37±0.02 21.79±1.34 1.06±0.08 1.15±0.07 0.31±0.01 48.74±4.11 4.20±0.35 0.14±0.01 22.24±1.86 24.68±1.52 53.08±4.47 |
|
0.47±0.03 14.59±1.23 6.32±0.54 0.48±0.03 0.32±0.02 0.59±0.04 20.51±1.94 1.08±0.09 1.17±0.11 0.25±0.02 50.17±4.52 3.95±0.31 0.11±0.01 22.18±1.85 23.60±2.20 54.22±4.84 |
|
Palmitic acid |
|
|
|||
|
Stearic acid |
|
|
|||
|
Arachidic acid |
|
|
|||
|
Behenic acid |
|
|
|||
|
Palmitoleic acid |
|
|
|||
|
Oleic acid |
|
|
|||
|
Vaccenic acid |
|
|
|||
|
Gondoic acid |
|
|
|||
|
Erucic acid |
|
|
|||
|
a linoleic acid |
|
|
|||
|
Linolenic acid |
|
|
|||
|
g linoleic aid |
|
|
|||
|
SFA |
|
|
|||
|
MUFA |
|
|
|||
|
PUFA |
|
|
|
|
C2 |
|
W2 |
|
R2 |
|
Miristic acid |
0.29±0.02 22.03±1.19 10.42±0.97 0.56±0.04 0.36±0.02 nd 16.78±1.21 1.52±0.12 1.08±0.99 0.49±0.03 42.05±3.21 4.15±0.78 0.27±0.04 33.66±2.24 19.87±2.35 46.47±4.03 |
|
0.23±0.02 17.37±1.21 6.99±0.05 0.56±0.03 0.48±0.02 0.27±0.02 18.51±0.99 1.13±0.11 0.69±0.05 0.54±0.03 49.59±3.56 3.12±0.27 0.52±0.04 25.63±1.33 21.14±1.2 53.23±3.87 |
|
0.27±0.01 18.65±1.67 8.19±0.07 0.40±0.01 0.33±0.01 0.42±0.02 17.87±1.28 1.20±0.11 0.78±0.09 0.27±0.01 48.01±4.23 3.13±0.24 0.15±0.01 27.84±1.77 20.87±1.52 51.29±2.02 |
|
Palmitic acid |
|
|
|||
|
Stearic acid |
|
|
|||
|
Arachidic acid |
|
|
|||
|
Behenic acid |
|
|
|||
|
Palmitoleic acid |
|
|
|||
|
Oleic acid |
|
|
|||
|
Vaccenic acid |
|
|
|||
|
Gondoic acid |
|
|
|||
|
Erucic acid |
|
|
|||
|
a linoleic acid |
|
|
|||
|
Linolenic acid |
|
|
|||
|
g linoleic aid |
|
|
|||
|
SFA |
|
|
|||
|
MUFA |
|
|
|||
|
PUFA |
|
|
Lines 502-503: What are "unripened fatty acids"?
Table 7: Should be "Tukey test".
Authors appreciate reviewer feedback. Both errors have been revised.
